# Burden of cystic and alveolar echinococcosis in the Kyrgyz Republic

**Gulnara Minbaeva**[1,2*], **Nelya Akhunbaeva**[2], **Kalis Nogoibaeva**[2], **Vera Toygombaeva**[1,2], **Kenesh Dzhusupov**[1], **Paul R. Torgerson**[3]

**1** International Higher School of Medicine, Bishkek, Kyrgyzstan, **2** Kyrgyz State Medical Academy named after I.K. Akhunbaeva, Bishkek, Kyrgyzstan, **3** Section of Veterinary Epidemiology Zurich University, Switzerland

\* minbaevagulnaramz82@gmail.com

## Abstract

The purpose of this study is to assess some aspects of the burden of cystic and alveolar echinococcosis in the Kyrgyz Republic. From 2015-2020 a total of 5568 cases of primary cystic echinococcosis (CE) were reported and a further 1008 recurrent cases. Over the same time period 880 primary cases of primary alveolar echinococcosis (AE) were reported and a further 343 recurrent cases. The estimated DALYs for CE was 7849, that is 1308 per annum or 19.5 per 100,000 per annum. For AE was there were 3809 DALYs, that is 634 per annum or 9.5 per 100,000 per annum. The proportion of fatal cases of AE was 3.6 times higher than from CE. The total economic damage in terms of purchasing power equivalent (PPE) from CE amounted to more than 10,5 million international US dollars. For AE it amounted to more than 5.1 million international US dollars. Most of the financial burden falls on the patients themselves and amounts to 68.4%, i.e., 1223 PPE dollars per case out of total spending 1615 PPE dollars per case. To reduce the social and economic burden of echinococcosis, it is necessary to timely identify the disease, prevent the development of complications and multiple lesions, and promote preventive chemotherapy where appropriate. A deep understanding of the etiology, epidemiology, clinical manifestations, and tactics of early diagnosis of various forms of echinococcosis is also important for timely detection, effective control, and, most importantly, the organization of preventive measures to combat this disease.

## Author summary

Echinococcosis is an important public health problem in Kyrgyzstan. From 2015 to 2020 a total of 5568 cases of primary cystic echinococcosis (CE) were reported and a further 1008 recurrent cases were reported to the Ministry of Health Kyrgyzstan. For alveolar echinococcosis (AE) 880 primary cases and a further 343 recurrent cases were reported. The burden of disease for CE was

**Data availability statement:** Some of the data is available in the manuscript. Sources of other data is referenced and available from the internet. Here the URL is given in the reference list [references 24–32].

**Funding:** This work was supported by the Swiss National Science Foundation (SNSF, grant agreement number 173131—"Transmission modelling of emergent echinococcosis in Kyrgyzstan") to PRT. URL: https://www.snf.ch/en. The funders did not play any role in the study design, data collection and analysis, preparation of the manuscript of decision to publish.

**Competing interests:** The authors have declared that no competing interests exist.

estimated at 1308 Disability Adjusted Life Years (DALYs) per annum, whilst for AE it was estimated to result in 634 DALYs per annum. In terms of purchasing power parity, the total economic costs of treatment for these disease amounted to more than $15 million over this 5 year period.

## Introduction

Echinococcosis is a highly pathogenic parasitic diseases in humans, characterized by space occupying lesions in the liver, lungs and other organs, which often lead to disability and occasional deaths [1]. A study from Armenia reported a case fatality rate of 1.29% [2]. This disease can cause substantial economic damage due to ill health. Globally, the WHO Reference Group on the Burden of Foodborne Disease Burden of Diseases estimated that echinococcosis is responsible for 19,300 deaths and approximately 871,000 disability-adjusted life years lost each year [3].

Cystic echinococcosis (CE) is a zoonotic parasitic disease with a global distribution caused by the larval stage of *Echinococcus granulosus sensu lato*. The distribution of *E. granulosus* is considered worldwide, with only a few areas such as Iceland, Ireland, and New Zealand believed to be free of autochthonous human CE. In the former Soviet Union and Eastern Europe, there was an initial increase in the number of observed cases from 1991 [4]. For example in Bulgaria the reported incidence increased from 2 per 100,000 in 1991 to 8 per 100,000 in 1996. The incidence started to decline in the first decade of this century, and by 2013 was 4 cases per 100,000 per year [5,6]. Likewise in Kazakhstan, the incidence of CE increased from 1.4 cases per 100,000 in 1991 to 5.9 cases per 100,000 in 2000 [7]. Thereafter, the incidence has been between 4.7 and 5.6 cases per 100,000 per year [8]. At least 270 million people (58% of the total population) are at risk of CE in Central Asia including areas of Mongolia, Kazakhstan, Kyrgyzstan, Tajikistan, Turkmenistan, Uzbekistan, Afghanistan, Iran, Pakistan and western China. The annual surgical incidence rate in part of central Asia has been estimated to be as high as 25–27 cases per 100,000 in some districts [9–11] with prevalences reaching 10% (range from 0.8 to 11.9%) documented in some Tibetan communities in western China [12] and for Kyrgyzstan by Paternoster et al. [13]. CE may not only cause severe disease and possible death in humans, but also results in economic losses from treatment costs, lost wages, and livestock-associated production losses. Globally, the annual cost of treating patients and losses in livestock production due to CE has been estimated at US$3 billion [1].

Alveolar echinococcosis (AE), caused by the larval stage of *Echinococcus multilocularis*, is confined to the northern hemisphere, in particular to North America, Central and Eastern Europe, Russia, Central Asia, Western China and Japan [14]. In humans, AE causes an infiltrating lesion primarily in the liver but in advanced cases metastasizing to other organs. It is usually a fatal disease if it remains untreated [15]. In some high endemic countries, the local prevalence of AE has been reported to be as high as 15% in some 7 villages in China [16] and up to 4% locally in Kyrgyzstan [17]. From 2014 to 2016, the surgical incidence of

AE on a country-level in Kyrgyzstan was 3.02 per 100,000 inhabitants per year, which is considered to be the highest country-wide incidence for AE worldwide [13]. Globally, approximately 10,500 new AE cases occur every year [14]. In most parts of the world, AE is considered as an emerging disease with increasing incidences over the last decades, although there appears to be a downward trend in China where the majority of cases are reported [14]. In Europe, a few hundred AE cases are diagnosed in every year with an increasing trend. In Switzerland, the yearly overall incidence of AE has significantly increased from 0.1 per 100,000 inhabitants between 1993–2000 to 0.26 per 100,000 in 2001–2005 [18]. Recent data suggests the incidence has increased further to an annual incidence of over 0.58 cases per 100,000 in 2021–2022 [19]. Kyrgyzstan is a lower middle income, mountainous country with 48% of the population employed in agriculture and one third of the population living below the poverty line. In recent years there is increasing evidence that AE and CE are becoming an increasing public health problem in central Asia, particularly in Kyrgyzstan where hundreds of cases are reported annually. The aims of this study were to estimate the burden of these disease in Kyrgyzstan and their monetary impact.

The global burden of AE and CE has been previously reported as approximately 690,000 and 184,000 DALYs respectively [3]. In Kyrgyzstan it has been previously estimated that the burden of AE is approximately 12,000 DALYs per annum or 50 DALYs per case. This assumed a case fatality rate approaching 100%. That for CE was approximately 3000 DALYs per annum, or 1.3 DALYs per cases with an assumed case fatality rate of 1% [20]. Treatment costs are highly variable depending on the local costs of medical treatment. For example, in Jordan, a lower middle income country, treatment costs for a case of CE was estimated at approximately US$500 in 2001 [21]. In contrast, in the United Kingdom, an upper income country the treatment costs of a single case of CE was estimated to be £7300 (aproximately US$10,000), also in 2001 [22]. Treatment costs of AE are higher and less well documented. In Switzerland, an upper income country, it was estimated that a single case of AE resulted in treatment costs of over US$100,000 [23].

## Methods

### Ethics

Ethical approval for the study was granted by the ethics committee of the Ministry of Health in Kyrgyzstan.This study was based on secondary data obtained from the "Automated Information System for Surveillance of Infectious and Parasitic Diseases" (AIS) of the Department of Disease Prevention and State Sanitary and Epidemiological Surveillance of the Ministry of Health of the Kyrgyz Republic for the period 2015–2020. All data were provided in a fully anonymized and aggregated form. The database contained no personal identifiers (including names, addresses, contact information, or individual identification numbers), making it impossible to identify individual patients or to link records to specific persons. All analyses and calculations were performed using aggregated anonymized data and were conducted in accordance with internationally accepted principles of bioethics. The analyses did not involve individual patient-level data and did not affect or target specific patients, thereby fully preserving confidentiality and anonymity.

### Patient data

The patient data were retrieved from the database of the "Automated Information System for Surveillance of Infectious and Parasitic Diseases" (AIS) of the Department of Disease Prevention and Public Health Surveillance of the Ministry of Health of the Kyrgyz Republic for 2015–2020 using the EPID platform [24]. Among others, relevant variables extracted for this study included date and clinical presentation of CE/AE, all surgical and anti-helminthic treatments in the in- and outpatient setting and long-term disability due to CE/AE or its treatment.

Data on the incidence and fatalities for CE and AE were analyzed from the AIS database for 2015–2020. Both diseases were cases based on clinical, pathological and histological confirmation of the lesion. Relapsed cases were those with the same diagnosis of CE or AE at the same anatomical site six or more months after discharge following treatment. Case reports included individual residential address, sex, age, occupation, diagnosis, i.e., CE, AE, CE relapse, or AE relapse.

The collected data were summed into the predefined 5 year age- and sex-categories, i.e., 0–4 yr, 5–9 yr, 10–14 yr, 15–19 yr, 20–24 yr, 25–29 yr, 30–34 yr, 35–39 yr, 40–44yr, 45–49yr, 50–54yr, 55–59yr, 60–64yr, 65–69 yr, 70–74 yr, 75–79 yr, 80–84yr, > 85 separately for males and females.

### Disability-adjusted life years (DALY) estimates

The disability-adjusted life year in its simplest terms can be considered a loss of healthy years of life and is a non-monetary measure of disease burden. It consists of two components: years lived with disability (YLDs) and years of life lost (YLLs). YLDs were calculated according to standard methods [3] using an incidence based approach for YLDs For patients diagnosed with echinococcosis a disability weight of 0.2, the same as for liver cancer (Dutch disability weight for disease free liver cancer) was assigned [1]. This disability weight was assumed for the length of the follow-up period. The WHO life table [25] was used to estimate YLLs for fatal cases

### Financial estimates

To assess the economic burden of CE and AE, an analysis of direct financial expenses of the state reimbursed by the Compulsory Medical Insurance Fund (CMIF). The data were officially presented by the CMIF of the Kyrgyz Republic, the cost of expenditure was determined in Kyrgyzstani som- (KGS). Additional expenses of the patients themselves during inpatient treatment of patients for 2015–2020 was carried out. Such costs include anti-relapse therapy, additional instrumental and laboratory tests and additional expenses of the patients themselves during the rehabilitation period which are not covered by the Medical Insurance Fund. Costs are presented in KGS, US $ and purchasing power equivalent $ (PPE), that is international dollars. The latter reflects the costs of goods and services in Kyrgyzstan compared to the costs of the same goods and services in the USA. Such goods and services are generally cheaper in Kyrgyzstan compared to the USA, thus $3.94 worth of goods and services in the USA would cost just $1 in Kyrgyzstan. Therefore the international dollar exchange rate used was 21.74 som to 1 US $ (2022 rate) [26]. The actual exchange rate used was 85.50 som to the US $, also the 2022 exchange rate (end of year) [27]. The costs of computer tomography, ultrasound examination, serological studies, histology and chemotherapy (albendazole) were sourced from [28–32].

### Statistical analysis

Comparisons of proportions was undertaken by the Fisher exact test, with p < 0.05 considered significant. Exact binomial confidence intervals were reported. All calculations were done in R [33].

## Results

The incidence of CE and AE from 2002 to 2022 among the population of the Kyrgyz Republic presented in Fig 1.

The data in Table 1. show that for the period 2015–2020 7,799 cases of echinococcosis received hospital treatment. For CE the proportion of cases that were primary cases was 84.7% on average with recurrent cases accounting for the remaining 15.3% of cases. The proportion of recurrent cases for AE was 28.0%. Of the total number of cases undergoing surgical treatment, the proportion of recurrences in AE was 1.8 times higher than in CE (p < 0.05).

Compared to 2015, there has been an increase in the incidence of recurrent cases rate since 2016, with a peak in 2018 of 4.4 per 100,000 and a subsequent downward trend by 2020 to 1.7 per 100,000.

An analysis of the reported cases of a 6-year period as a whole showed that for the period 2015–2020 the case fatality rate for AE was 3.6 times higher than for CE. For AE 36 cases from 1,223 registered cases had a fatal outcome (2.9%) whilst for CE there were where 55 fatalities from 6,576 registered cases (0.9%) (p < 0.0001).

For children under 14 years of age there were three fatal cases of AE from 30 registered case (10%), which was not significantly different from the adult case fatality rate of 33 fatal cases from 1,193 registered cases (p = 0.055). For CE

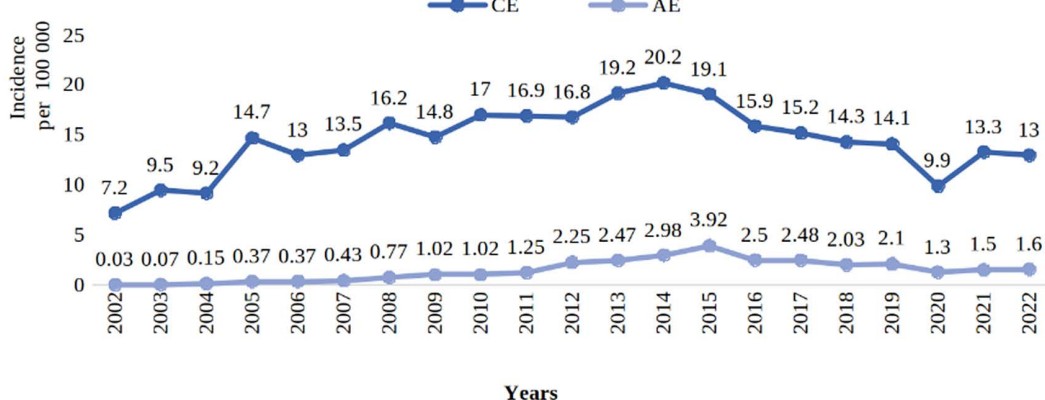

**Fig 1. The incidence of new cases CE and AE among the population of the Kyrgyz Republic, 2001-2022.** Incidence calculated from the numbers of cases notified to the Ministry of Health.

**Table 1. The ratio of patients with primary n=6,448 and recurrent n=1,351 CE and AE treated in hospitals, (n=7,799) and the number of fatal cases (n=91), Kyrgyz Republic, 2015-2020.**

|  | Cystic echinococcosis | | |
|---|---|---|---|
|  | **Primary** | **Recurrent** | **Proportion Recurrent (95% CI)** |
| 2015 | 1,134 | 34 | 2.9% (2.0% to 4.0%) |
| 2016 | 963 | 213 | 21.1% (19.5% to 22.6%) |
| 2017 | 942 | 225 | 22.3% (20.7% to 23.8%) |
| 2018 | 906 | 250 | 24.8% (23.2% to 26.3%) |
| 2019 | 970 | 194 | 19.2% (17.6% to 20.7%) |
| 2020 | 653 | 92 | 9.1% (7.5% to 10.6%) |
| **Total** | **5,568** | **1,008** | **15.3% (13.7% to 16.8%)** |
|  | Alveolar echinococcosis | | |
|  | **Primary** | **Recurrent** | **Proportion Recurrent (95% CI)** |
| 2015 | 235 | 87 | 25.4% (24.0% to 26.7%) |
| 2016 | 150 | 53 | 15.5% (14.1% to 16.8) |
| 2017 | 149 | 73 | 21.3% (19.9% to 22.6%) |
| 2018 | 121 | 66 | 19.2% (17.8% to 20.5%) |
| 2019 | 137 | 42 | 12.2% (10.8% to 13.5%) |
| 2020 | 88 | 22 | 6.4% (5.0% to 7.7%) |
| **Total** | **880** | **343** | **28.0% (26.6% to 29.3%)** |

there were no fatal cases reported among the 1,136 registered pediatric cases, compared to 55 deaths from the 4,940 registered cases in adults (p<0.0001, Tables 1 and 2).

The total DALY from CE was 7,849 and from AE was 3,809 over 6 years (Table 3) which is a mean of 1,308 and 634 DALYs respectively per year. Each deceased patient with CE lost a mean of 48 YLLs and whilst fatal AE cases lost a mean of 46 YLLs. The annual burden of disease per 100,000 population for CE was 19.5 DALYs and that for AE was 9.5 DALYs. For CE, it was estimated that there was approximately 1.2 DALYs per case, whilst for AE it was approximately 3.1 DALYs per case.

**Table 2. Case fatality rates of CE and AE cases treated in hospitals.**

| | Fatal cases from CE | | | Fatal cases from AE | | |
|---|---|---|---|---|---|---|
| | Number of cases | Fatalities | Case fatality rate (95% CI) | Number of cases | Fatalities | Case fatality rate (95% CI) |
| 2015 | 1,168 | 8 | 0.7% (0.3% to 1.4%) | 322 | 5 | 1.5% (0.5% to 3.5%) |
| 2016 | 1,176 | 5 | 0.4% (0.1% to 1.0%) | 203 | 3 | 1.4% (0.03% to 4.2%) |
| 2017 | 1,167 | 13 | 1.1% (0.5% to 1.9%) | 122 | 7 | 3.1% (1.2% to 6.3%) |
| 2018 | 1,156 | 14 | 1.2% (0.6% to 2.0%) | 187 | 10 | 5.3% (2.5% to 9.6%) |
| 2019 | 1,166 | 11 | 0.9% (0.4% to 1.6%) | 179 | 7 | 3.9% (1.5% to 7.8%) |
| 2020 | 745 | 4 | 0.5% (0.1% to 1.3%) | 110 | 4 | 3.6% (1.0% to 9.0%) |
| **Total** | **6,576** | **55** | **0.8% (0.6% to 1.0%)** | **1,223** | **36** | **2.9% (2.0% to 4.0%)** |

Table 3 presents the age stratified DALYs estimated during the study period. This includes the relative contributions from YLLs and YLDs. As can be seen from the data in the Kyrgyz Republic, the CMIF for 2015–2020 reimbursed financial expenses to healthcare organizations for inpatient treatment with primary and recurrent CE and AE for 7,799 patients. In general, over five years, the total cost of treatment with additional expenses for patients amounted to ≈ 66.9 million som (KGS), that is 782,711 US dollars.

The average cost of one treated (direct state costs) case of CE is 8,376 KGS and alveolar echinococcosis is 9,680 KGS. The average cost of CE and AE was 8,580 KGS per patient, that is, ≈ 100 US dollars. When converted to the international exchange rate against the US dollar, which was 21.74 at the time of calculation (https://en.wikipedia.org/wiki/International_dollar), the cost to the government per patient would be the purchasing power equivalent (PPE) of 394 US dollar (Table 4).

The total financial costs for CE amounted to 55 082 046 million KGS or 644,234 US $. Direct government expenditures for the treatment of AE amounted to 11,839,735 million KGS or 138 476 US $.

As shown in Table 5, the direct costs of the patients themselves, in comparison with the direct costs of the state, are three times higher. For CE the state spends a total of $98 per patient and the patients' own costs are $311, which is 3.2 times more. For alveolar echinococcosis, the state spends a total of $100 per patient, and patients' own costs are $420, which is 4.2 times more. According to our calculations, about 73% of the costs are borne by the patients themselves, which has a direct negative impact on the economic situation of the family. It should be noted that our calculations do not take into account indirect costs such as transportation, additional food, patient care, accommodation and temporary disability.

The total economic damage in terms of PPE from CE should have amounted to more than 10.5 million international US dollars. For AE should have amounted to more than 5.1 million US dollars. The total economic damage in terms of PPE from CE and AE should have amounted to more than 14.4 US million international dollars. Most of the financial burden falls on the patients themselves and amounts to 68.4%, i.e., 1223 PPE dollars out of total spending 1615 PPE dollars per case.

## Discussion

The World Health Organization (WHO) has included CE in its official list of 18 neglected tropical diseases. In addition to this group, WHO has defined CE as belonging to a subgroup of seven endemic or "neglected zoonotic diseases". Echinococcosis is common where there is poverty and a dependence on livestock. These are indicators of the quality of social and financial capital, low resilience and close proximity of people and their animals contribute to the transmission of this disease [34]. In the context of echinococcosis surveillance and control, mass screening surveys provide valuable data that will help reduce the medical, social and economic burden by ensuring early detection and rapid treatment of cases and in humans.

**Table 3. Age distribution of fatal cases of CE and AE and the DALYs estimate of reduction in life expectancy from due to premature death (AE), Kyrgyz Republic, 2015-2020.**

| Cystic Echinococcosis | | | | | | | | | |
|---|---|---|---|---|---|---|---|---|---|
| Age | Residual life expectance | Cases n=6,576 | Deaths n=55 | YLLs | YLDs fatal | YLDs non fatal | DALYs non fatal | DALYs fatal | Total DALY |
| 0-4 | 87.1 | 103 | 0 | 0.0 | 0.0 | 9.4 | 9.4 | 0.0 | 9.4 |
| 5-9 | 82.6 | 223 | 0 | 0.0 | 0.0 | 81.4 | 81.4 | 0.0 | 81.4 |
| 10-14 | 77.6 | 810 | 0 | 0.0 | 0.0 | 138.5 | 138.5 | 0.0 | 138.5 |
| 15-19 | 72.6 | 342 | 8 | 580.8 | 2.5 | 102.8 | 686.1 | 581.2 | 1269.3 |
| 20-24 | 67.6 | 37 | 5 | 338.1 | 0.4 | 11.7 | 350.1 | 339.0 | 690.1 |
| 25-29 | 62.7 | 23 | 1 | 62.7 | 0.3 | 7.3 | 70.3 | 63.6 | 133.3 |
| 30-34 | 57.7 | 458 | 7 | 404.0 | 0.5 | 205.7 | 610.1 | 404.9 | 1017.2 |
| 35-39 | 52.8 | 359 | 5 | 263.8 | 0.4 | 157.4 | 421.6 | 264.7 | 687.7 |
| 40-44 | 47.8 | 521 | 3 | 143.5 | 0.5 | 236.2 | 380.2 | 144.6 | 525.0 |
| 45-49 | 42.9 | 1,312 | 5 | 214.7 | 0.5 | 596.0 | 811.1 | 215.7 | 1028.1 |
| 50-54 | 38.1 | 752 | 8 | 304.8 | .4 | 322.3 | 627.5 | 306.4 | 935.8 |
| 55-59 | | 784 | 3 | 100.0 | 0.5 | 356.1 | 456.6 | 101.0 | 557.9 |
| 60-64 | 28.7 | 300 | 3 | 86.0 | 0.4 | 132.0 | 218.5 | 88.3 | 305.8 |
| 65-69 | 24.1 | 456 | 2 | 48.2 | 0.4 | 170.8 | 219.4 | 48.8 | 268.4 |
| 70-74 | 19.8 | 58 | 0 | 0.0 | 0.0 | 26.4 | 26.4 | 0.0 | 26.4 |
| 75-79 | 15.7 | 38 | 5 | 78.3 | 0.5 | 15.0 | 93.8 | 78.5 | 174.3 |
| 80-84 | 11.96 | 0 | 0 | 0 | 0 | 0 | 0 | 0.0 | 0.0 |
| 85< | 7.05 | 0 | 0 | 0 | 0 | 0 | 0 | 0.0 | 0.0 |
| Total | 811.0 | 6,576.0 | 55 | 2,624.8 | 7.1 | 2,569.0 | 5,200.9 | 2,636.5 | 7,849 |

| Alveolar Echinococcosis | | | | | | | | | |
|---|---|---|---|---|---|---|---|---|---|
| Age | Residual Life Expectancy | AE cases n=1223 | AE deaths n=36 | YLL AE | YLDs fatal AE | YLDs non fatal AE | DALYs non fatal AE | DALYs fatal | Total DALY |
| 0-4 | 87.1 | 1 | 1 | 87.1 | 0.2 | 0.0 | 87.2 | 87.2 | 174.5 |
| 5-9 | 82.6 | 29 | 2 | 165.2 | 0.6 | 8.0 | 173.8 | 165.8 | 339.5 |
| 10-14 | 77.6 | 43 | 0 | 0.0 | 0.0 | 60.8 | 60.8 | 0.0 | 60.8 |
| 15-19 | 72.6 | 36 | 1 | 72.6 | 0.3 | 11.2 | 84.1 | 72.9 | 157.0 |
| 20-24 | 67.6 | 57 | 0 | 0.0 | 0.0 | 14.3 | 14.3 | 0.0 | 14.3 |
| 25-29 | 62.7 | 172 | 1 | 62.7 | 0.3 | 58.5 | 121.5 | 63.0 | 184.5 |
| 30-34 | 57.7 | 112 | 4 | 230.8 | 1.1 | 30.8 | 262.8 | 232.0 | 494.7 |
| 35-39 | 52.8 | 200 | 9 | 474.8 | 4.0 | 84.9 | 563.8 | 478.8 | 1042.6 |
| 40-44 | 47.8 | 146 | 1 | 47.8 | 0.3 | 44.6 | 92.8 | 48.1 | 140.9 |
| 45-49 | 42.9 | 252 | 2 | 85.9 | 0.8 | 105.5 | 192.2 | 86.7 | 278.9 |
| 50-54 | 38.1 | 86 | 2 | 76.2 | 0.7 | 29.7 | 106.6 | 76.9 | 183.5 |
| 55-59 | 33.3 | 28 | 2 | 66.7 | 0.7 | 9.5 | 76.9 | 67.4 | 144.3 |
| 60-64 | 28.7 | 31 | 7 | 200.6 | 1.8 | 6.0 | 208.4 | 202.4 | 410.8 |
| 65-69 | 24.1 | 0 | 2 | 48.2 | 0.0 | 0.0 | 48.2 | 48.2 | 96.5 |
| 70-74 | 19.8 | 13 | 2 | 39.5 | 0.4 | 2.4 | 42.3 | 40.0 | 82.3 |
| 75-79 | 15.7 | 17 | 0 | 0.0 | 0.0 | 3.7 | 3.7 | 0 | 3.7 |
| 80-84 | 11.96 | 0.0 | 0.0 | 0.0 | 0.0 | 0.0 | 0.0 | 0.0 | 0.0 |
| 85< | 7.05 | 0.0 | 0.0 | 0.0 | 0.0 | 0.0 | 0.0 | 0.0 | 0.0 |
| Total | 811.0 | 1,223.0 | 36.0 | 1,658.1 | 11.3 | 469.8 | 2,139.2 | 1,669.4 | 3,809 |

**Table 4. Direct government costs (in KGS) for the treatment of patients with various forms of CE, AE and unspecified diagnoses in hospitals across the country, 2015-2020. Data from the health insurance fund.**

**Cystic echinococcosis**

| Site of primary lesion | Per Case | | | Number of cases | Total Costs | | |
|---|---|---|---|---|---|---|---|
| | KGS | Exchange rate 85.50 | International dollar pp exchange rate = 21.74 | | KGS | Exchange rate 85.50 | International dollar pp exchange rate = 21.74 |
| Liver | 7,108 | 83 | 327 | 1,620 | 11,514,960 | 134,678 | 529,667 |
| Lung | 9,104 | 106 | 419 | 360 | 3,277,440 | 38,333 | 150,756 |
| Bone | 7,211 | 84 | 332 | 120 | 865,320 | 10,121 | 39,803 |
| Other local./multip. | 8,483 | 99 | 390 | 2,730 | 23,158,590 | 270,861 | 1,065 253 |
| Unspecified | 9,316 | 109 | 429 | 1,746 | 16,265,736 | 190,243 | 748,194 |
| **Total CE** | **8,376** | **98** | **385** | **6,576** | **55,082,046** | **644,234** | **2,533,673** |

**Alveolar echinococcosis**

| Site of primary lesion | KGS | Exchange rate 85.50 | International dollar pp exchange rate = 21.74 | Number of cases | KGS | Exchange rate 85.50 | International dollar pp exchange rate = 21.74 |
|---|---|---|---|---|---|---|---|
| Liver | 8,121 | 95 | 374 | 241 | 1,957,161 | 22,891 | 90,026 |
| Other local./multip. | 10,123 | 118 | 466 | 940 | 9,515,620 | 111,294 | 437,701 |
| Unspecified | 8,737 | 102 | 402 | 42 | 366,954 | 4 292 | 16,879 |
| **Total AE** | **9,680** | **113** | **445** | **1,223** | **11 839 735** | **138 476** | **544,606** |
| **Total government cost** | **8,580** | **100** | **394** | **7,799** | **66921,781** | **782,711** | **3,078,279** |

**Table 5. Additional Direct costs of patients (in KGS) with various forms of diagnoses CE and AE in hospitals across the country, n = 3619, 2015-2020.**

| | CE | CE | AE | AE | Total |
|---|---|---|---|---|---|
| | N = 1 | N = 6,576 | N = 1 | N = 1,223 | N = 7,799 |
| Complete blood count, complete urine test, biochemical tests (creatinine, bilirubin, blood clotting), electrocardiography | 2,000 | 13,152,000 | 2,000 | 2,446,000 | 15,598,000 |
| Additional expenses: Computed tomography, ultrasound, serological studies | 5,950 | 39,127,200 | 11,250 | 13,758,750 | 68,558,050 |
| Blood plasma 1 dose | 0 | 0 | 1,700 | 2,079,100 | 7,106,000 |
| Histological analyzes | 1,990 | 13,086,240 | 1,990 | 2,433,770 | 15,520,010 |
| Antibiotic resistance tests | 1,368 | 8 995 968 | 1,368 | 1,673,064 | 10,669,032 |
| Cost for surgery | 11,300 | 74,308,800 | 13,600 | 16,632,800 | 97,742 700 |
| Costs for 3 months of anti-relapse chemotherapy (drug Albendazole) | 3,982 | 26,185,632 | 3,982 | 4,869,986 | 31,055,618 |
| **Total cost patients** | **26,590** | **174,855,840** | **35,890** | **43,893,470** | **246,249,410** |
| **Exchange rate 85.50** | **311** | **2,045,098** | **420** | **513,374** | **2,880,110** |
| **International dollar pp exchange rate = 21.74** | **1,223** | **8,043,047** | **1,651** | **2,019,019** | **11,327,020** |

The annual incidence of CE in Kyrgyzstan, based on hospital records, increased from 5.4 cases/100,000 in 1991–14 cases/100,000 in 1998 [10], and then to 20 cases/100,000 in 2014 (the present study). Between 1995 and 2011, human AE increased from <3 cases per year to >60 cases per year and then to over 200 per year by 2015 [13,35]. The prevalence of CE also increased in major livestock species such as sheep with a doubling of reported prevalence in some areas [10]. An increase in the number of deaths from echinococcosis was observed in 2017, 2018 and 2019. The case fatality rate of 2.9%, is higher for AE compared to the 0.8% for CE AE would usually have a high case fatality rate, although

concomitant treatment with albendazole following liver resection does improve the prognosis [15,36]. The low case fatality rate reported in the present study may be due to the under reporting of deaths with AE, which could be registered as liver failure or neoplasia. Also, the large increases in the reported cases of AE over the last 20 years indicate an evolving epidemic. AE is a chronic disease and death may occur several years after diagnosis, so AE cases reported in the present study may result in fatalities in the future. Alternatively, with increasing use of albendazole following surgical treatment may be dramatically decreasing the case fatality rate.There was a substantial decrease in the number of reported cases and deaths due to CE and AE in 2020 (Fig 1 and Tables 1 and 2). This was in the same year as the start of the COVID 19 pandemic and hence this decrease in reported incidence of echinococcosis may be due to the health services having to manage the large numbers of COVID 19 patients and thus fewer resources were availbale to investigate and treat cases of suspected echinococcosis.

The sharp rise in treatment costs—particularly by 1.4 times noted in the year 2020 compared to 2017—reflects both increased awareness and reporting of the disease as well as potentially rising prices for medical care and pharmaceuticals. It should be noted that in the sum of CE and AE diseases, the share of costs for the treatment of AE of other organs unspecified more than 9.5 million KGS, i.e., ≈ 111.3 thousand US$. However, the clinical manifestations of CE are more diverse, and the economic burden of cases with multiple localization more than 23 million KGS, i.e., 270 thousand US$. This indicates a late detection of this form of clinical manifestation of CE.

The high number of disability-adjusted life years – DALYs and deaths demonstrates not only the deadliness of these diseases but also indicate additional social and economic burdens on families and society as a whole. Furthermore, many cases, especially those in remote areas where there is poor medical services infrastructure, may go undiagnosed and unreported. For example, in the remote Alay valley in Kyrgyzstan, a prevalence of over 4% for AE in the population has been reported [17]. Such unreported cases will add significantly to the burden of disease. The DALY estimated in the present report are somewhat lower for AE than previously estimated by Coulotte et al. [20]. This is because in the previous study, AE was assumed to have a high case fatality rate and estimated YLLs on the assumption that most diagnosed cases of AE would eventually be fatal cases. The present study estimates the YLLs only on reported fatal cases and will be an underestimate if a substantial number of the AE cases would indeed be fatal, either due to misdiagnosis or deferred deaths at a later date. The burden of disease of CE of 1.2 DALYs per case is in line with previous studies [3], but the 3.1 DALYs per case for AE is somewhat lower than other studies and is likely to be due to the reported case fatality rate as described above.

## Public health challenges

Echinococcosis is emblematic of larger public health challenges facing the Kyrgyz Republic, particularly the integration of effective zoonotic disease management with overall health system strengthening. The persistence and escalation of echinococcosis burden underscores the need for a One Health approach, which links human, animal, and environmental health strategies cohesively.

The increasing economic and social burden of CE and AE highlights the need for enhanced public health policy interventions in the Kyrgyz Republic. Enhanced funding for health services, especially in rural and under-served areas, is critical. Implementing widespread screening and early diagnosis programs, particularly in high-risk areas, could significantly reduce the incidence and severity of the disease. Public health campaigns aimed at educating the population about the transmission, dog population management and prevention of echinococcosis can also play a crucial role. Additionally, strengthening the healthcare infrastructure to better manage and treat cases will mitigate the long-term economic impact and improve patient outcomes.

## Limitations of the study

One of the main limitations of the study is the potential under-reporting of cases and deaths associated with echinococcosis, which may lead to an underestimation of the disease's true burden. The study authors may miss unreported cases,

especially in rural areas where there is a lack of medical reporting and resources, as well as AE deaths may be reported as cancer, cirrhosis, etc. Also, the economic estimates do not account for indirect costs such as lost productivity and long-term disability, which could significantly increase the calculated burden. Furthermore, the monetary value of deaths has not been estimated as this is a controversial area in health economics and was one of the reasons the DALY was adopted to assess the burden of disease.

### Future research perspectives

Future studies should aim to incorporate factors mentioned above to provide a more comprehensive view of the burden of echinococcosis. Such research should focus on improving disease surveillance and reporting mechanisms to capture all cases and outcomes related to echinococcosis, thus providing a more accurate and comprehensive data set.

Further research is necessary to explore the genetic and molecular aspects of *Echinococcus* species prevalent in the region to understand the transmission dynamics better and develop targeted treatments. Additionally, studies assessing the impact of climate change on the epidemiology of echinococcosis could provide insights into future trends and help in formulating adaptive strategies.

## Conclusion

This comprehensive analysis of CE and AE in the Kyrgyz Republic has underscored a significant escalation in both the economic and social burdens of these diseases over the period studied. Our findings illuminate the growing incidence of these conditions and their profound impact on public health systems, individual patients, and broader societal dynamics.

Our study calls for a robust response to echinococcosis due to substantial costs and disease burden. This should emphasize the importance of sustained financial investment in health infrastructure, the enhancement of epidemiological surveillance, and the mobilization of community education efforts. By addressing echinococcosis with a comprehensive and proactive approach, the Kyrgyz Republic can better manage and eventually reduce the burden of this severe disease, leading to improved public health outcomes and enhanced economic stability.

## Acknowledgments

We also wish to acknowledge the the MHIF staff who provided data from the reporting forms.

## Author contributions

**Conceptualization:** Gulnara Minbaeva, Paul R. Torgerson.

**Data curation:** Gulnara Minbaeva, Nelya Akhunbaeva, Kalis Nogoibaeva, Vera Toygombaeva, Kenesh Dzhusupov.

**Formal analysis:** Gulnara Minbaeva.

**Investigation:** Gulnara Minbaeva, Kalis Nogoibaeva, Vera Toygombaeva, Kenesh Dzhusupov.

**Methodology:** Gulnara Minbaeva, Nelya Akhunbaeva, Vera Toygombaeva, Kenesh Dzhusupov.

**Resources:** Nelya Akhunbaeva, Kalis Nogoibaeva, Vera Toygombaeva.

**Supervision:** Vera Toygombaeva, Paul R. Torgerson.

**Validation:** Gulnara Minbaeva.

**Visualization:** Gulnara Minbaeva.

**Writing – original draft:** Gulnara Minbaeva.

**Writing – review & editing:** Gulnara Minbaeva, Paul R. Torgerson.

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
