## [Decision Letter · Decision Letter 0]

26 Oct 2025

ECONOMIC AND SOCIAL BURDEN OF CYSTIC AND ALVEOLAR ECHINOCOCCOSIS IN THE KYRGYZ REPUBLIC

Dear Dr. Torgerson,

Thank you for submitting your manuscript to PLOS Neglected Tropical Diseases. After careful consideration, we feel that it has merit but does not fully meet PLOS Neglected Tropical Diseases's publication criteria as it currently stands. Therefore, we invite you to submit a revised version of the manuscript that addresses the points raised during the review process.

Please submit your revised manuscript within 60 days Dec 25 2025 11:59PM. If you will need more time than this to complete your revisions, please reply to this message or contact the journal office at plosntds@plos.org. Please include the following items when submitting your revised manuscript:

We look forward to receiving your revised manuscript.

Kind regards,

Majid Fasihi Harandi, PhD

Academic Editor

Eva Clark

Section Editor

Shaden Kamhawi

co-Editor-in-Chief

Paul Brindley

co-Editor-in-Chief

**Journal Requirements:**

At this stage, the following Authors/Authors require contributions: Nelya Akhunbaeva, Vera Toygombaeva, Kenesh Dzhusupov, and Kalis Nogoybaeva. Please ensure that the full contributions of each author are acknowledged in the "Add/Edit/Remove Authors" section of our submission form.

4) Thank you for including an Ethics Statement for your study. Please include:

i) A statement that formal consent was obtained (must state whether verbal/written) OR the reason consent was not obtained (e.g. anonymity). NOTE: If child participants, the statement must declare that formal consent was obtained from the parent/guardian.].

5) Please upload all main figures as separate Figure files in .tif or .eps format. For more information about how to convert and format your figure files please see our guidelines:

6) Please provide a complete Data Availability Statement in the submission form, ensuring you include all necessary access information or a reason for why you are unable to make your data freely accessible. If your research concerns only data provided within your submission, please write "All data are in the manuscript and/or supporting information files" as your Data Availability Statement.

7) Please amend your detailed Financial Disclosure statement. This is published with the article. It must therefore be completed in full sentences and contain the exact wording you wish to be published.

8) Please send a completed 'Competing Interests' statement, including any COIs declared by your co-authors. If you have no competing interests to declare, please state "The authors have declared that no competing interests exist". Otherwise please declare all competing interests beginning with the statement "I have read the journal's policy and the authors of this manuscript have the following competing interests"

**Reviewers' Comments:**

Reviewer's Responses to Questions

**Key Review Criteria Required for Acceptance?**

**Methods**

-Are the objectives of the study clearly articulated with a clear testable hypothesis stated?

-Is the study design appropriate to address the stated objectives?

-Is the population clearly described and appropriate for the hypothesis being tested?

-Is the sample size sufficient to ensure adequate power to address the hypothesis being tested?

-Were correct statistical analysis used to support conclusions?

-Are there concerns about ethical or regulatory requirements being met?

Reviewer #1: Is the 5 year follow-up sufficient to monitor mortality caused by echinococcosis?

Reviewer #2: See the attached file

Reviewer #3: The study addresses an important question and uses appropriate national datasets (AIS and CMIF) but lacks methodological clarity. Objectives are relevant yet not expressed as a clear, testable hypothesis. Case definitions for CE, AE, and relapse are insufficiently described, and it is unclear whether relapse cases were excluded from incidence denominators.

Statistical analysis is limited to basic proportion tests; trend or regression analyses would better support conclusions. DALY calculations require clarification of approach (incidence vs prevalence), duration, disability weight sources, and uncertainty estimation.

Currency conversion methods (nominal vs PPP) are confusing.

Ethical approval is noted

**Results**

-Does the analysis presented match the analysis plan?

-Are the results clearly and completely presented?

-Are the figures (Tables, Images) of sufficient quality for clarity?

Reviewer #1: Figure 1: primary and relapse cases both included? Please, indicate in legend.

Table 1 : the meaning of the column “proportion” is not clear. The title mentions lethality?? Please, indicate how this proportion is calculated

Table 3: indicate meaning of WHO GHE. Possibly indicate in column titles the formula used.

Table 4: meaning of column “proportion”? Add formulas as well?

Reviewer #2: See the attached review file

Reviewer #3: Results are broadly aligned with the study’s objectives but inconsistently presented. Some tables contain formatting and labelling errors, varying totals, and unclear confidence intervals. Figures and captions are mismatched in date ranges, and abbreviations (WHO GHE) are not always defined.

While descriptive patterns appear plausible (rising incidence, higher recurrence in AE, substantial costs), the analyses are not statistically robust and lack uncertainty reporting. Data presentation requires careful editing for accuracy and clarity.

**Conclusions**

-Are the conclusions supported by the data presented?

-Are the limitations of analysis clearly described?

-Do the authors discuss how these data can be helpful to advance our understanding of the topic under study?

-Is public health relevance addressed?

Reviewer #1: Line 204: please discuss a bit more the impact of COVID 19 outbreak of echinococcosis detection?

End of discussion: why not mentioning vaccination in control tools of echinococcosis?

Conclusion: too long with many repetitions from the discussion

Reviewer #2: See the attached review file

Reviewer #3: The conclusions reflect the data directionally but overstate evidential strength. Claims of increasing incidence and costs are not statistically demonstrated, and uncertainty is not addressed. Limitations such as under-reporting and diagnostic misclassification are acknowledged.

Public-health relevance is well noted, particularly regarding surveillance and One Health approaches, but the discussion can be more concise.

**Editorial and Data Presentation Modifications?**

Reviewer #1: - missing dot line 45

- line 81 and others: indent

- line 133: verb is missing

- line 190: dollarsout

- line 201: detection and rapid treatment of cases and in humans

Reviewer #2: See the attached review file

Reviewer #3: Line(s) Comment / Correction

33 “senso lata” → sensu lato

34 “E. granulosis” → E. granulosus

36 “dramatically increased” — specify since when

36–39 Add a reference for the increase.

53 “100.000” → “100,000” (apply throughout)

76–77 Remove repeated phrase: “Ministry of Health of the Kyrgyz Republic for 2015–2020”.

102–103 You introduce “CMIF” but do not use it later — either use consistently or remove.

103 “som” → “Kyrgyzstani Som”.

113 “figure 1” → Figure 1 (capitalise).

Figure 1 Caption states 2001–2022, but figure shows 2002–2022 — correct discrepancy.

117 “Table 1.” → Table 1 (remove full stop).

118–120 Sentence beginning “For CE, primary cases accounted…” — rewrite for clarity and fix punctuation.

122–124 Sentence beginning “Compared to 2015 and 2020…” — remove the first mention of 2020; redundant and confusing.

126–127 “Table 1 The ratio…” → “Table 1. The ratio of patients with primary n = 6,448 and recurrent n = 1,351 CE and AE treated in hospitals (n = 7,799) and the number of fatal cases (n = 91), Kyrgyz Republic, 2015–2020.” (also apply consistent formatting throughout).

129–132 “What dynamics?” — please rewrite for clarity and fix punctuation.

135 Use CE consistently after first definition; do not revert to full term mid-section.

137 Correct punctuation around p-value (e.g. p < 0.05).

141 “table 3” → Table 3 (also throughout).

143–144 Remove duplication: “The, the annual burden”.

Table 3 Include definitions for abbreviations in caption (e.g. WHO GHE). Ensure “(AE)” appears in the correct section heading.

153–154 If abbreviations (e.g. CMIF) are introduced, use them consistently thereafter.

155 “7 799” → “7,799”.

167 Spell out “three”.

169 “alveolar echinococcosis” → “AE”.

168–170 Be consistent in formatting of “US $” or “USD” throughout.

Table 4 The table reads poorly due to column-label formatting. Clarify all columns and headings, especially “Exchange rate 85.50”. Move exchange rates and PPP factor to caption. Place total counts first for readability. Explain the meaning of the 95% CI column — why does it have both a value and a range? Reformat and relabel for clarity.

Table 5 Provide references for cost values in the Methods, not within the table.

186–188 Replace conditional “should have amounted” with “has amounted”. Also change “10,5 million” → “10.5 million”.

190 “dollarsout” → “dollars out”.

208–209 Clarify “low according/in relation to what?” — add comparison or context.

231 “Coulotte et al. [14]” → “Coulotte et al. [14].” (add final period).

236–239 Corrected sentence:

“The annual incidence of CE in Kyrgyzstan, based on hospital records, increased from 5.4 cases/100,000 in 1991 to 14 cases/100,000 in 1998 [5], and then to 20 cases/100,000 in 2014 (the present study). Between 1995 and 2011, human AE increased from < 3 cases per year to > 60 cases per year, and then to over 200 per year by 2015 [11, 18].”

243 Ensure all subtitles are correctly formatted (consistent style and hierarchy).

258–265 The “Strengths of the study” paragraph is unnecessary; consider removing.

— Sample sizes vary between 7,789 and 7,799 in different sections — reconcile all totals.

**Summary and General Comments**

Reviewer #1: This is a very interesting manuscript, well written and presented, deserving publication in PLOS neglected tropical diseases.

Reviewer #2: See the attached review file

Reviewer #3: This manuscript estimates the epidemiological and economic burden of cystic (CE) and alveolar echinococcosis (AE) in Kyrgyzstan using national surveillance (AIS) and insurance (CMIF) data for 2015–2020, and derives DALYs and treatment costs.

The topic is important and policy-relevant, and the dataset appears substantial. However, there are significant methodological and reporting issues that must be addressed before the results can be considered reliable.

PLOS authors have the option to publish the peer review history of their article (what does this mean? ). If published, this will include your full peer review and any attached files.). If published, this will include your full peer review and any attached files.

**Do you want your identity to be public for this peer review?** For information about this choice, including consent withdrawal, please see our For information about this choice, including consent withdrawal, please see our Privacy Policy ..

Reviewer #1: **Yes:** Marcotty TanguyMarcotty Tanguy

Reviewer #2: **Yes:** Amer Al-JawabrehAmer Al-Jawabreh

Reviewer #3: **Yes:** Mahbod EntezamiMahbod Entezami

**Figure resubmission:**
---

## [Decision Letter · Decision Letter 1]

28 Jan 2026

BURDEN OF CYSTIC AND ALVEOLAR ECHINOCOCCOSIS IN THE KYRGYZ REPUBLIC

Dear Dr. Torgerson,

Thank you for submitting your manuscript to PLOS Neglected Tropical Diseases. After careful consideration, we feel that it has merit but does not fully meet PLOS Neglected Tropical Diseases's publication criteria as it currently stands. Therefore, we invite you to submit a revised version of the manuscript that addresses the points raised during the review process.

* A letter that responds to each point raised by the editor and reviewer(s). You should upload this letter as a separate file labeled 'Response to Reviewers '. This file does not need to include responses to any formatting updates and technical items listed in the 'Journal Requirements' section below.'. This file does not need to include responses to any formatting updates and technical items listed in the 'Journal Requirements' section below.

* A marked-up copy of your manuscript that highlights changes made to the original version. You should upload this as a separate file labeled 'Revised Manuscript with Track Changes '.'.

* An unmarked version of your revised paper without tracked changes. You should upload this as a separate file labeled 'Manuscript '.'.

We look forward to receiving your revised manuscript.

Kind regards,

Majid Fasihi Harandi, PhD

Academic Editor

Eva Clark

Section Editor

Shaden Kamhawi

co-Editor-in-Chief

Paul Brindley

co-Editor-in-Chief

**Journal Requirements:**

1) Please provide an Author Summary. This should appear in your manuscript between the Abstract (if applicable) and the Introduction, and should be 150-200 words long. The aim should be to make your findings accessible to a wide audience that includes both scientists and non-scientists. Sample summaries can be found on our website under Submission Guidelines:

2) Thank you for including an Ethics Statement for your study. Please include:

i) A statement that formal consent was obtained (must state whether verbal/written) OR the reason consent was not obtained (e.g. anonymity). NOTE: If child participants, the statement must declare that formal consent was obtained from the parent/guardian.].

**Reviewers' comments:**

**Key Review Criteria Required for Acceptance?**

**Methods**

-Are the objectives of the study clearly articulated with a clear testable hypothesis stated?

-Is the study design appropriate to address the stated objectives?

-Is the population clearly described and appropriate for the hypothesis being tested?

-Is the sample size sufficient to ensure adequate power to address the hypothesis being tested?

-Were correct statistical analysis used to support conclusions?

-Are there concerns about ethical or regulatory requirements being met?

Reviewer #1: yes

Reviewer #2: See attached file

Reviewer #3: The authors have adequately addressed previous queries regarding the definitions used for recurrence and the methodology for calculating DALYs. The objectives remain clear and the study design is appropriate for a retrospective analysis of national surveillance data.

**Results**

-Does the analysis presented match the analysis plan?

-Are the results clearly and completely presented?

-Are the figures (Tables, Images) of sufficient quality for clarity?

Reviewer #1: yes

Reviewer #2: See attached file

Reviewer #3: The presentation of the results is currently unacceptable for publication due to severe formatting errors. Table 4 is unintelligible in its current form, the headers contain unprofessional typographical errors (e.g., "Internationao", "doller") and the columns are misaligned, making it impossible to verify the economic data. Additionally, Table 1 lacks consistent decimal formatting (mixing commas and points).

**Conclusions**

-Are the conclusions supported by the data presented?

-Are the limitations of analysis clearly described?

-Do the authors discuss how these data can be helpful to advance our understanding of the topic under study?

-Is public health relevance addressed?

Reviewer #1: yes

Reviewer #2: See attached file

Reviewer #3: The conclusions are broadly supported by the data, but the clarity is undermined by inconsistent terminology. The authors interchange "mortality" (deaths per population) and "case fatality rate" (deaths per case) when comparing AE and CE. Precise terminology is required here to accurately convey the comparative severity of the diseases.

**Editorial and Data Presentation Modifications?**

Reviewer #1: yes. Minor editorial issues.

Reviewer #2: I have attached the R1 PDF file with my comments in blue at the end of the file

Reviewer #3: The edits have been noted in the review docuement

**Summary and General Comments**

Reviewer #1: yes

Reviewer #2: See attached file

Reviewer #3: While the authors have addressed several methodological queries raised in the first round, the revised manuscript is not in a publishable state. The authors stated that specific corrections were made, yet the manuscript file provided still contains the original errors. Furthermore, Tables 4 and 5 contain severe formatting errors and typos (e.g. "Internationao", "doller").

PLOS authors have the option to publish the peer review history of their article (what does this mean? ). If published, this will include your full peer review and any attached files.). If published, this will include your full peer review and any attached files.

**Do you want your identity to be public for this peer review?** For information about this choice, including consent withdrawal, please see our For information about this choice, including consent withdrawal, please see our Privacy Policy ..

Reviewer #1: **Yes:** Marcotty TanguyMarcotty Tanguy

Reviewer #2: **Yes:** Prof. Amer Al-JawabrehProf. Amer Al-Jawabreh

Reviewer #3: **Yes:** Dr. Mahbod EntezamiDr. Mahbod Entezami

**Figure resubmission:**
---

## [Decision Letter · Decision Letter 2]

23 Feb 2026

Dear Prof Torgerson,

We are pleased to inform you that your manuscript 'BURDEN OF CYSTIC AND ALVEOLAR ECHINOCOCCOSIS IN THE KYRGYZ REPUBLIC' has been provisionally accepted for publication in PLOS Neglected Tropical Diseases.

Best regards,

Majid Fasihi Harandi, PhD

Academic Editor

Eva Clark

Section Editor

Shaden Kamhawi

co-Editor-in-Chief

Paul Brindley

co-Editor-in-Chief

Reviewer's Responses to Questions

**Key Review Criteria Required for Acceptance?**

**Methods**

-Are the objectives of the study clearly articulated with a clear testable hypothesis stated?

-Is the study design appropriate to address the stated objectives?

-Is the population clearly described and appropriate for the hypothesis being tested?

-Is the sample size sufficient to ensure adequate power to address the hypothesis being tested?

-Were correct statistical analysis used to support conclusions?

-Are there concerns about ethical or regulatory requirements being met?

Reviewer #2: As this is R2, please see the attached pdf file with my feedback in blue.

Reviewer #3: The study objectives are clearly articulated, focusing on the social and economic burden of echinococcosis in Kyrgyzstan. The study design uses secondary data from the Automated Information System for Surveillance of Infectious and Parasitic Diseases, which is appropriate for a retrospective burden-of-disease analysis. The population is clearly described and appropriate for calculating incidence and DALYs.The authors have clarified that the statistical analysis was conducted using R and have provided specific significance thresholds as requested. Ethical requirements have been sufficiently addressed with a detailed statement on data anonymisation and approval by the Ministry of Health.

**Results**

-Does the analysis presented match the analysis plan?

-Are the results clearly and completely presented?

-Are the figures (Tables, Images) of sufficient quality for clarity?

Reviewer #2: As this is R2, please see the attached pdf file with my feedback in blue.

Reviewer #3: The analysis presented follows the described methodology for YLL, YLD, and economic cost estimation. The results are now more clearly presented following the correction of typographical and formatting errors in the tables. The authors have rectified the typos in Table 4 and aligned the columns to ensure the economic data is now intelligible. Decimal formatting is now consistent throughout Table 1.

**Conclusions**

-Are the conclusions supported by the data presented?

-Are the limitations of analysis clearly described?

-Do the authors discuss how these data can be helpful to advance our understanding of the topic under study?

-Is public health relevance addressed?

Reviewer #2: As this is R2, please see the attached pdf file with my feedback in blue.

Reviewer #3: The conclusions regarding the high economic and social burden are supported by the presented data, showing a high patient-borne cost of roughly 68.4%. The limitations, including potential under-reporting and the exclusion of indirect costs, are clearly described. The authors effectively discuss the relevance of these data for public health policy and the necessity of a One Health approach in the region. Terminology has been tightened to correctly distinguish between case fatality rate and mortality.

**Editorial and Data Presentation Modifications?**

Reviewer #2: As this is R2, please see the attached pdf file with my feedback in blue.

The track-changes word file showed several minor editorial errors which can be taken care of easily during the publication process.

Reviewer #3: (No Response)

**Summary and General Comments**

Reviewer #2: As this is R2, please see the attached pdf file with my feedback in blue.

Reviewer #3: This study provides a significant and novel assessment of the economic and social burden of CE and AE in the Kyrgyz Republic, a region with some of the highest incidences globally. The strengths of the study lie in its use of comprehensive national surveillance data and the application of DALY and PPE metrics to quantify disease impact. In this third round of revision, the authors have successfully improved the scholarship and presentation of the manuscript by correcting formatting errors and inconsistent terminology. The manuscript is now in a publishable state and contributes valuable health-economic insights into neglected zoonotic diseases.

PLOS authors have the option to publish the peer review history of their article (what does this mean? ). If published, this will include your full peer review and any attached files.). If published, this will include your full peer review and any attached files.

**Do you want your identity to be public for this peer review?** For information about this choice, including consent withdrawal, please see our For information about this choice, including consent withdrawal, please see our Privacy Policy ..

Reviewer #2: **Yes:** Prof. Amer Al-JawabrehProf. Amer Al-Jawabreh

Reviewer #3: **Yes:** Mahbod EntezamiMahbod Entezami

---

## [Editor Report · Acceptance letter]

Dear Prof Torgerson,

We are delighted to inform you that your manuscript, "BURDEN OF CYSTIC AND ALVEOLAR ECHINOCOCCOSIS IN THE KYRGYZ REPUBLIC," has been formally accepted for publication in PLOS Neglected Tropical Diseases.

Best regards,

Shaden Kamhawi

co-Editor-in-Chief

Paul Brindley

co-Editor-in-Chief
